# Complex carbohydrate utilization by gut bacteria modulates host food consumption

Kristie B. Yu [1] ✉, Celine Son[1], Ezgi Özcan [1,3], Anisha Chandra[1], Jorge Paramo[2], Andrew Varghese[1], Alicia Roice[1], Delanie Finnigan[1], Franciscus Chandra[1], Anna Novoselov[1], Sabeen A. Kazmi [1], Gregory R. Lum[1], Arlene Lopez-Romero [2], Jonathan B. Lynch [1,4] & Elaine Y. Hsiao [1,2] ✉

The gut microbiota interacts with dietary nutrients and can modify host feeding behavior, but underlying mechanisms remain poorly understood. Gut bacteria digest complex carbohydrates that the host cannot digest and liberate metabolites that serve as energy sources and signaling molecules. Here, we use a gnotobiotic mouse model to examine how gut bacterial fructose polysaccharide metabolism influences host intake of diets containing these carbohydrates. Two *Bacteroides* species ferment fructans with different glycosidic linkages: *B. thetaiotaomicron* ferments levan with β2-6 linkages, whereas *B. ovatus* ferments inulin with β2-1 linkages. We find that mice eat relatively more diet containing the carbohydrate that their gut bacteria cannot ferment compared to the fermentable ones: mice colonized with *B. thetaiotaomicron* consume more inulin diet, while mice colonized with *B. ovatus* consume more levan diet. Knockout of bacterial fructan utilization genes attenuates this difference, whereas swapping the fermentation ability of *B. thetaiotaomicron* to inulin confers increased consumption of levan diet. Bacterial fructan fermentation and host feeding behavior are associated with neuronal activation in the arcuate nucleus of the hypothalamus. These results reveal that bacteria nutrient metabolism modulates host food consumption through sensing of differential energy extraction, which contributes to our understanding of determinants of food choice.

Eating behavior that supports intake of a well-balanced diet is essential for health. The gut microbiota interacts directly with dietary nutrients to regulate host metabolism and energy storage from food[1,2]. Particular microbes specialize in metabolizing different types of nutrients to support their cellular respiration and to ultimately survive in a competitive gut microbial ecosystem[3]. This has led some to hypothesize that gut microbes may manipulate host feeding behavior to promote their own fitness[4]. Conversely, the host must have mechanisms to keep microbial loads in check due to nutrient and physical constraints of the gut, such as bacterial proteins produced during stationary growth phase acting on host anorexigenic circuits to suppress appetite[5].

Growing evidence supports a role for the gut microbiota in modulating host macronutrient preference, particularly for palatable diets containing high levels of fat or sugar. Transplanting microbiota from obese donor mice into lean recipient mice lowered preference for high-fat diets, which was correlated with deficits in dopamine signaling in the nucleus accumbens[6,7]. Supplementation with inulin, a

[1]Department of Integrative Biology and Physiology, University of California, Los Angeles, Los Angeles, CA, USA. [2]UCLA Goodman-Luskin Microbiome Center, Vatche and Tamar Manoukian Division of Digestive Diseases, Department of Medicine, David Geffen School of Medicine, Los Angeles, CA, USA. [3]Present address: School of Nutrition and Food Sciences, Louisiana State University Agricultural Center, Baton Rouge, LA, USA. [4]Present address: Department of Biological Chemistry, Johns Hopkins School of Medicine, Baltimore, MD, USA. ✉e-mail: kbyu@mednet.ucla.edu; ehsiao@g.ucla.edu

prebiotic fiber that alters the gut microbiota, increased mouse preference for high-fat diet over high-sugar diet[8]. Conversely, treating mice with antibiotics to deplete gut bacteria led them to overconsume palatable high-sugar diets and exhibit increased activation in mesolimbic brain regions[9]. These studies provide foundational proof-of-concept that alterations in the gut microbiome are sufficient to influence host feeding behavior. However, they do not determine whether there is a causal relationship between selective nutrient utilization by gut bacteria and host dietary preference, which can be difficult to tease apart when palatable macronutrients such as fat or sugar themselves strongly influence host brain circuits[6,7,9].

Therefore, studying host dietary preference for less palatable macronutrients such as protein and fiber may better elucidate mechanisms for microbial modulation of host dietary preference. In fruit flies deprived of essential amino acids, gut bacteria suppressed host protein appetite by producing lactate-derived metabolites or amino acids that are detected by gut expression of the neuropeptide CNMamide[10,11]. In mice, colonization with microbiota from wild rodents with different foraging strategies conferred differential food preferences, where those colonized with herbivore-associated microbiota preferred diets with a higher protein to carbohydrate ratio diet than those colonized with omnivore- and carnivore-associated microbiota[12]. Herbivore-associated microbiota showed higher abundance of aromatic amino acid synthesis genes than the other diets correlating with plasma tryptophan levels[12], suggesting that bacterial metabolism can alter plasma essential amino acid levels in mice as in flies[10,11]. These studies provide mechanistic insight into how gut microbial metabolism modulates protein preference.

Gut microbial metabolism of fiber, which refers to complex carbohydrates that are non-digestible by the host[1,2,13], generates short chain fatty acids (SCFAs) that can reduce host appetite through gut-brain pathways[14-17]. Despite these known effects on satiety, there are no studies to date that focus on dietary fiber preference. To address this, we leverage well-characterized pathways for bacterial fiber metabolism to investigate a causal mechanism by which gut microbes can influence host food consumption. We focus on species from the genus *Bacteroides*, prevalent human-derived gut which are critical for digesting complex carbohydrates via genetically encoded polysaccharide utilization loci (PUL)[18] to produce the SCFAs acetate and propionate[14]. *Bacteroides thetaiotaomicron* and *Bacteroides ovatus* express PUL variants that restrict their metabolism to fructose-containing polysaccharides called fructans with different glycosidic linkages: *B. thetaiotaomicron* ferments levan-type fructans with β2-6 linkages, whereas *B. ovatus* ferments inulin-type fructans with β2-1 linkages[19,20]. We use these two phylogenetically related and genetically tractable bacteria in a gnotobiotic mouse model to dissect mechanisms by which bacterial utilization of complex carbohydrates impacts host consumption of diets that differ only in carbohydrate type. We further manipulate bacterial PULs to examine causal effects of bacterial fructan utilization on host diet consumption and neuronal activation in the arcuate nucleus of the hypothalamus. Results from this study reveal that selective fructan utilization by gut bacteria shapes host food consumption behavior and advance mechanistic understanding of how the gut microbiota impacts host feeding behavior.

## Results
### Colonization with fructan-utilizing bacteria promotes host consumption of non-fermentable fructan diet
The gut microbiome has the capacity to modify host appetite and preference for diets that vary in macronutrient composition[6,7,9-11,15,16]. To determine whether differential fructan utilization by gut bacteria alters host consumption of diets that vary only in fructan source, we began by selecting *B. ovatus* and *B. thetaiotaomicron* based on their reported selectivity in degrading inulin and levan[19], respectively, which are fructans that differ only in their glycosidic linkages (Fig. 1a). We

generated custom mouse diets containing either 10% inulin or levan purified for dietary formulation ("dietary levan", see Methods) as the only carbohydrate source (Supplementary Table S1). The diets have matching caloric content (63% fat and 17% protein by weight), so that any differential effects would be due to bacterial fructan utilization. As expected, *B. thetaiotaomicron* grew in minimal media containing 0.5% of either pure levan or dietary levan, but not inulin as its sole carbohydrate source; *B. ovatus* grew in minimal media containing inulin, but not pure levan (Fig. 1b). It also exhibited partial growth on dietary levan, suggesting the presence of residual mono- or oligosaccharides generated from the bulk purification process (Fig. 1b). Consistent with this, *B. thetaiotaomicron* grew better in minimal media containing 5% levan diet (LD) than inulin diet (ID), while *B. ovatus* exhibited only a slight growth advantage in minimal media containing ID than LD (Fig. 1c and Supplementary Fig. 1a). The observed decrease in bacterial count after 12-16 h likely reflects the exhaustion of the limited amount of fructan present. The SCFA acetate is a major end product of bacterial carbohydrate fermentation[13]. We confirmed in vitro that *B. thetaiotaomicron* produced more acetate on LD than ID, while *B. ovatus* produced more acetate on ID than LD (Fig. 1d). Therefore, we refer in subsequent sections to LD as a fermentable diet and ID as a non-fermentable diet to *B. thetaiotaomicron*, and LD as a non-fermentable diet and ID as a fermentable diet to *B. ovatus*.

Fiber fermentation by the gut microbiota contributes an estimated 10% of total energy requirements in humans[21]. We therefore hypothesized that differential bacterial fructan utilization would alter host fructan diet consumption through host sensing of differential energy extraction. To test this, we mono-colonized mice with wild-type *B. thetaiotaomicron* or *B. ovatus*, fed them ID or LD in sequence, and measured their food consumption (Fig. 1e). To minimize potential confounds of diet novelty, mice were habituated to both ID and LD for 7 days prior to colonization, and diet order was counterbalanced in each cohort. Throughout the experiment, there were no differences in mouse body weight or fecal bacterial load based on diet or colonization status (Supplementary Fig. 1b, c). Mice mono-colonized with *B. thetaiotaomicron* exhibited higher average daily intake of their non-fermentable ID compared to fermentable LD (Fig. 1f), which was mainly driven by the order of diet exposure: mice exposed to their non-fermentable fructan first (*Bt* NF→F) ate less of the fermentable fructan compared to the non-fermentable fructan (Fig. 1g). While no significant differences in diet intake were seen when considering all mice colonized with *B. ovatus* (Fig. 1f), the same pattern of altered intake was apparent when groups were examined by diet order: only when exposed to the non-fermentable diet first (*Bo* NF→F) did mice consume less of their fermentable ID than non-fermentable LD (Fig. 1g). To further gain insight into whether this difference in sequential diet intake holds in the presence of both diets, mice were subjected at the end of the experimental paradigm to both diets within the home cage, and overnight intake of each diet was quantified (Fig. 1e). Consistent with results from average daily intake, mice colonized with *B. thetaiotaomicron* ate more of their non-fermentable ID than fermentable LD in the overnight assay (Fig. 1h), which was mainly driven by the cohort exposed to the non-fermentable fructan first (*Bt* NF→F) (Fig. 1i). A similar pattern of relatively increased consumption of the non-fermentable diet, in a diet order-dependent manner, was seen in mice colonized with *B. ovatus*, but it did not reach statistical significance for the overnight assay (Fig. 1h, i). The stronger phenotypes with *B. thetaiotaomicron* may be due to its clearer growth bias in LD than ID, whereas *B. ovatus* exhibited substantial baseline growth in LD that was only modestly surpassed in ID (Fig. 1c). However, there is no correlation between average daily diet intake and overnight diet intake "preference" (proxy for non-fermentable diet intake minus fermentable intake) in either group (Supplementary Fig. 1d). To gain insight into whether increased consumption of the non-fermentable diet reflects a change from baseline, we colonized germ-free (GF) mice with

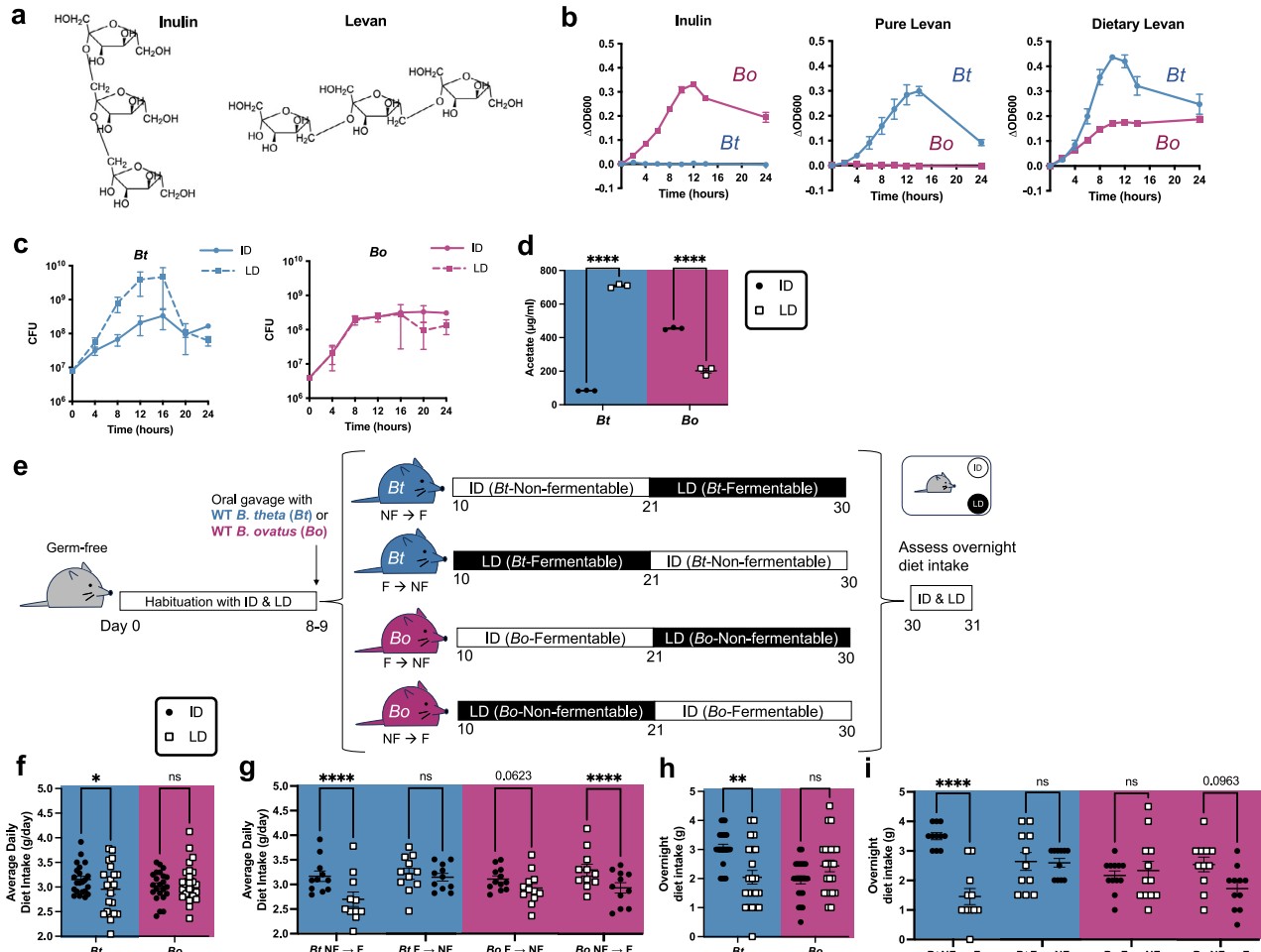

**Fig. 1 | Colonization with differential fructan-utilizing bacteria increases host consumption of diets containing the non-fermentable fructan. a** Fructan chemical structures. Inulin has β2-1 linkages between fructose subunits, while levan has β2-6 linkages between fructose subunits. **b** Growth curves of *B. thetaiotaomicron (Bt)* and *B. ovatus (Bo)* in minimal media containing 0.5% inulin, pure levan, or dietary levan. Each dot represents mean of 3 biological replicates, error bars represent SEM. OD600 optical density at 600 nm. **c** Growth curves of *Bt* and *Bo* in minimal media containing 5% inulin diet (ID) or levan diet (LD). Each dot represents mean of 3 biological replicates, error bars represent SEM. CFU colony forming unit. **d** Acetate produced in supernatants of bacteria grown for 24 h in minimal media containing 5%ID or LD. (*n* = 3 each). *p*-values (from left to right) = <0.0001, <0.0001. ANOVA interaction *p*-value < 0.0001. **e** Experimental schematic for sequential feeding post-colonization. Germ-free mice were habituated with both diets for 7 days before being colonized with *Bt* or *Bo*. The mice ate one diet *ad libitum* for 11 days and then the other diet for 10 days. The mice were then provided both diets in their cage overnight. **f** Average daily diet intake post-colonization for mice

colonized with *Bt* and *Bo*. *p*-values (from left to right) = 0.0301, 0.5389. ANOVA interaction *p*-value = 0.0159. **g** Average daily diet intake post-colonization for mice colonized with *Bt* and *Bo* by diet order. *p*-values (from left to right) = <0.0001, 0.8281, 0.0623, <0.0001. ANOVA interaction *p*-value < 0.0001. **h** Overnight diet intake from days 30–31 for mice colonized with *Bt* and *Bo*. *p*-values (from left to right) = 0.0030, 0.1981. ANOVA interaction *p*-value = 0.0009. **i** Overnight diet intake from days 30–31 for mice colonized with *Bt* and *Bo* by diet order. *p*-values (from left to right) = <0.0001, 0.9999, 0.9798, 0.0963. ANOVA interaction *p*-value < 0.0001. Error bars for (**d**, **f**–**i**) represent mean ± SEM. Data from (**f**–**i**) are combined from three independent experiments. For (**f** and **h**), *n* = 22 for *Bt*, 23 for *Bo*. For (**g** and **i**), *n* = 11 for *Bt* NF → F, 11 for *Bt* F → NF, 12 for *Bo* F → NF, 11 for *Bo* NF → F. For (**d**, **f**–**i**), 2-way ANOVA with matched column measures, comparing means across rows, and Sidak's corrections were performed. ns = *p*-value > 0.10; * = *p*-value < 0.05; **** = *p*-value < 0.0001. ID inulin diet, LD levan diet, F fermentable diet, NF non-fermentable diet.

*Bt, Bo*, or conventional microbiota (CONV), and presented them both diets concurrently, without any prior habituation to the diets (Fig. 1e). When presented with both diets at the same time, all three groups consumed more LD than ID (Supplementary Fig. 1f). This may be due to the presence of residual mono- or oligosaccharides generated from the bulk purification process for dietary levan (Fig. 1b). Together, these results show that gut microbial fructan utilization leads to a change in food consumption behavior: mice colonized with *B. thetaiotaomicron* consumed relatively less of the fermentable LD than non-fermentable ID when exposed in NF→F sequence (Fig. 1f–i), which may be driven by post-ingestive feedback of more energy extracted from the fermentable diet, despite a baseline preference for LD diet (Supplementary Fig. 1f). This aligns with our hypothesis that host sensing of differential

energy extraction by bacteria would alter host fructan diet consumption.

## Bacterial fructan utilization drives host consumption of non-fermentable diet

Fructan PUL components *susC* and *susD* determine binding specificity and utilization of inulin[22,23] and levan[19,20] (Supplementary Fig. 2a). To determine whether gut bacterial fructan utilization is necessary or sufficient to alter host diet intake, we used allelic exchange[24] to manipulate the fructan utilization ability of *B. thetaiotaomicron* and *B. ovatus*. We deleted the *susC*- and *susD*-like genes in the fructan PUL of *B. thetaiotaomicron* (BT1762-1763) and *B. ovatus* (BACOVA_04504-04505) to generate *Bt*^ΔsusCD^ and *Bo*^ΔsusCD^ (Supplementary Fig. 2b), which

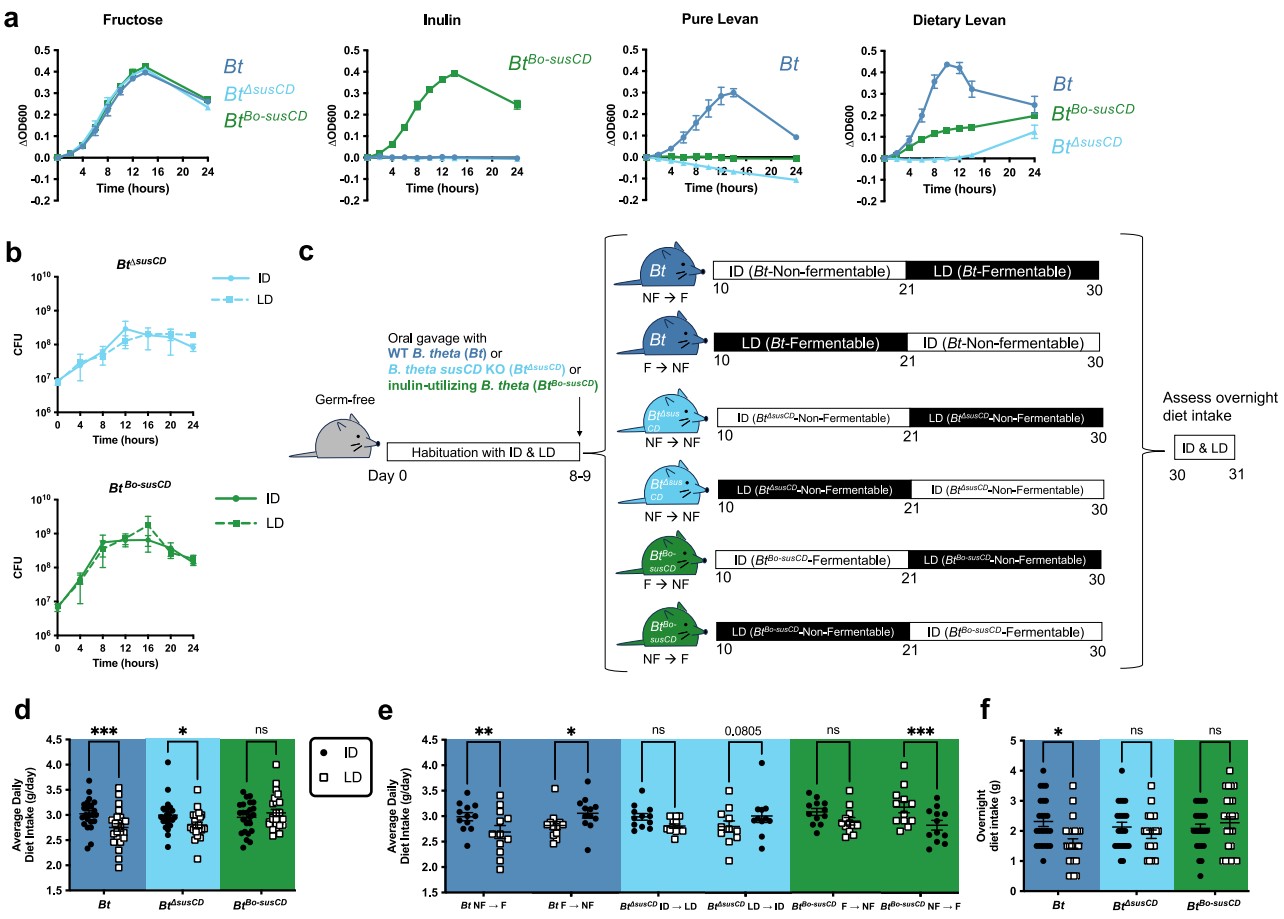

**Fig. 2 | Bacterial fructan utilization drives host consumption of diets containing the non-fermentable fructan. a** Growth curves of wild-type *B. thetaiotaomicron (Bt)*, mutant *B. thetaiotaomicron* lacking *susCD* (*Bt^ΔsusCD*), and inulin-utilizing *B. thetaiotaomicron* (*Bt^Bo-susCD*) in minimal media containing 0.5% fructose, inulin, levan (pure), or levan (purified for dietary formulation). Each dot represents mean of 3 biological replicates, error bars represent SEM. OD600 optical density at 600 nm. **b** Growth curves of *Bt*, *Bt^ΔsusCD*, and *Bt^Bo-susCD* in minimal media containing 5% inulin diet (ID) or levan diet (LD). Each dot represents mean of 3 biological replicates, error bars represent SEM. CFU colony forming unit. **c** Experimental schematic for sequential feeding post-colonization. **d** Average daily diet intake post-colonization for mice colonized with *Bt*, *Bt^ΔsusCD*, and *Bt^Bo-susCD*. *p*-values (from left to right) = 0.0002, 0.0115, 0.4467. ANOVA interaction *p*-value = 0.0004. **e** Average daily diet

intake post-colonization for mice colonized with *Bt*, *Bt^ΔsusCD*, and *Bt^Bo-susCD* by diet order. *p*-values (from left to right) = 0.0016, 0.0174, 0.1318, 0.0805, 0.1244, 0.0001. ANOVA interaction *p*-value < 0.0001. **f** Overnight diet intake from days 30–31 for mice colonized with *Bt*, *Bt^ΔsusCD*, and *Bt^Bo-susCD*. *p*-values (from left to right) = 0.0220, 0.7600, 0.8595. ANOVA interaction *p*-value = 0.0556. Error bars for (**d**–**f**) represent mean ± SEM. Data from (**d**–**f**) are combined from three independent experiments. For (**d** and **f**), *n* = 24 for *Bt*, 23 for *Bt^ΔsusCD*, and 24 for *Bt^Bo-susCD*. For (**e**), *n* = 12 for *Bt* NF → F, 12 for *Bt* F → NF, 12 for *Bt^ΔsusCD* ID → LD, 11 for *Bt^ΔsusCD* LD → ID, 12 for *Bt^Bo-susCD* F → NF, and 12 for *Bt^Bo-susCD* NF → F. For (**d**–**f**), 2-way ANOVA with matched column measures, comparing means across rows, and Sidak's corrections were performed. ns = *p*-value > 0.10; * = *p*-value < 0.05; ** = *p*-value < 0.01; *** = *p*-value < 0.001. ID inulin diet, LD levan diet, F fermentable diet, NF non-fermentable diet.

were no longer able to grow in minimal media containing inulin or levan (Figs. 2a and 3a). To generate an isogenic strain of *B. thetaiotaomicron* that can degrade inulin but not levan, we replaced BT1762-1763 with BACOVA_04504-04505, and deleted the levan-specific glycoside hydrolase BT1760 (Supplementary Fig. 2c). The resulting strain *Bt^Bo-susCD* exhibited the fructan specificity of *B. ovatus*, growing in minimal media containing inulin but not levan (Fig. 2a). We further confirmed that the genetic modifications resulted in functional differences in production of SCFAs[14]. *B. thetaiotaomicron* produced high levels of acetate when grown in its fermentable LD, which was substantially diminished in *Bt^ΔsusCD*, and reversed in *Bt^Bo-susCD*, which produced high levels of acetate when grown in ID (Supplementary Fig. 2d). This aligned with results from *B. ovatus*, where the wild-type strain produced high acetate levels when grown in its fermentable ID, but *Bo^ΔsusCD* failed to produce acetate in either ID or LD. These patterns were not seen for propionate (Supplementary Fig. 2e), suggesting that

acetate is the dominant SCFA produced by fructan utilization for these particular bacterial species.

To determine how bacterial fructan utilization impacts host feeding behavior, we mono-colonized mice with wild-type *B. thetaiotaomicron*, deletion mutant *Bt^ΔsusCD,* or gene replacement mutant *Bt^Bo-susCD*, fed them ID or LD in sequence for 10 days each, and measured their food consumption (Fig. 2c). We confirmed that *Bt^ΔsusCD* exhibits poor growth in minimal media containing LD compared to wild-type *B. thetaiotaomicron* (Fig. 2b and Supplementary Fig. 3a). *Bt^Bo-susCD* exhibited strong growth in minimal media containing inulin and ID, but like *B. ovatus* (Fig. 1b, c), it had no growth with pure levan and intermediate growth with dietary levan and LD (Fig. 2a, b and Supplementary Fig. 3a). Mice showed no differences in body weight or bacterial load based on diet or colonization status (Supplementary Fig. 3b, c), indicating that fructan utilization is not required for successful gut colonization. As previously observed (Fig. 1f, g), mice colonized with wild-

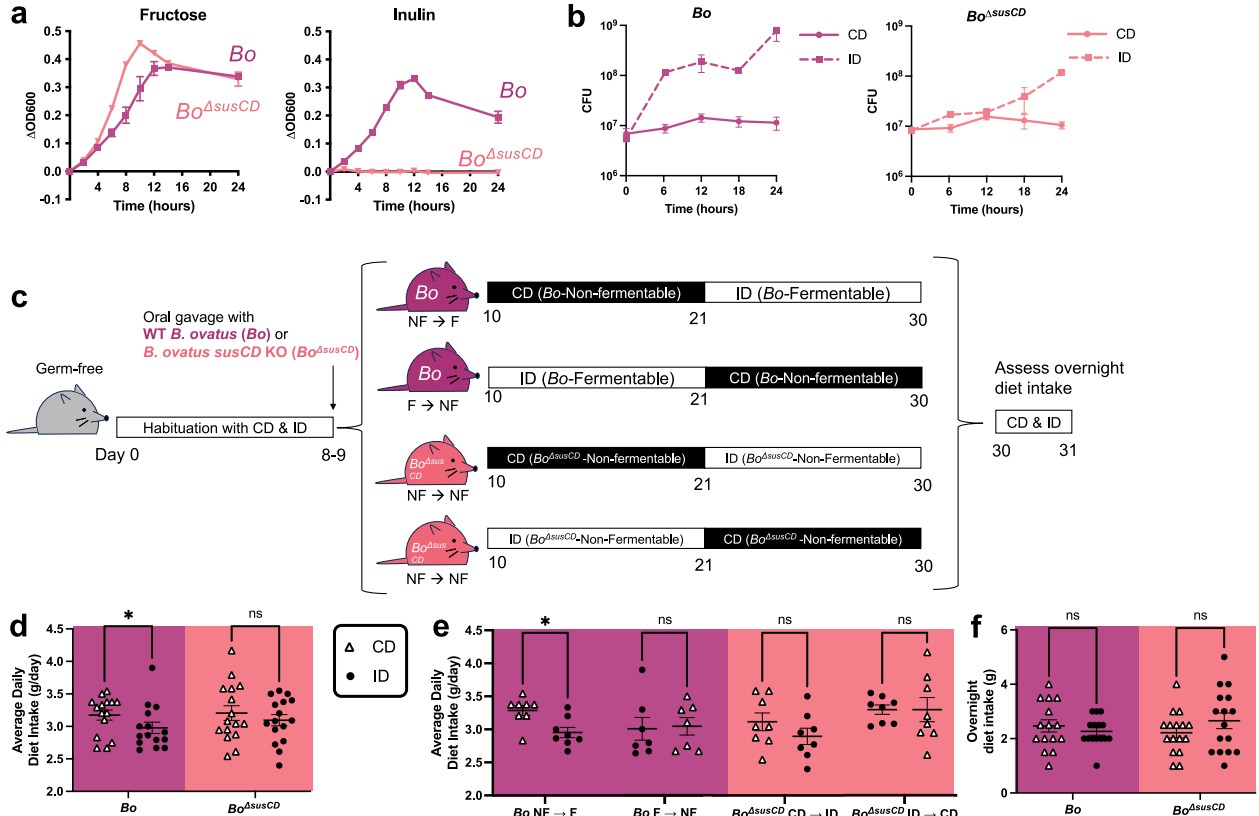

**Fig. 3 | Colonization with fructan-utilizing gut bacteria increases host consumption of fructan-free diet. a** Growth curves of wild-type *B. ovatus (Bo)* and *B. ovatus susCD* KO (*Bo^{ΔsusCD}*) in minimal media containing 0.5% fructose or inulin. Each dot represents mean of 3 biological replicates, error bars represent SEM. OD600 optical density at 600 nm. **b** Growth curves of *Bo* and *Bo^{ΔsusCD}* in minimal media containing 5% cellulose diet (CD) or inulin diet (ID). Each dot represents mean of 3 biological replicates, error bars represent SEM. CFU colony forming unit. **c** Experimental schematic for sequential feeding post-colonization. **d** Average daily diet intake post-colonization for mice colonized with *Bo* and *Bo^{ΔsusCD}*. *p*-values (from left to right) = 0.0402, 0.3021. ANOVA interaction *p*-value = 0.4461. **e** Average daily diet intake post-colonization for mice colonized with *Bo* and *Bo^{ΔsusCD}* by diet order.

*p*-values (from left to right) = 0.0114, 0.9948, 0.1490, >0.9999. ANOVA interaction *p*-value = 0.0971. **f** Overnight diet intake from days 30–31 for mice colonized with *Bo* and *Bo^{ΔsusCD}*. *p*-values (from left to right) = 0.7558, 0.2584. ANOVA interaction *p*-value = 0.1337. Error bars for (**d–f**) represent mean ± SEM. Data from (**d–f**) are combined from two independent experiments. For (**d** and **f**), *n* = 15 for *Bo*, 16 for *Bo^{ΔsusCD}*. For (**e**), *n* = 8 for *Bo* NF → F, 7 for *Bo* F → NF, 8 for *Bo^{ΔsusCD}* CD → ID, and 8 for *Bo^{ΔsusCD}* ID → CD. For (**d–f**), 2-way ANOVA with matched column measures, comparing means across rows, and Sidak's corrections were performed. ns = *p* > 0.10; * = *p*-value < 0.05. CD cellulose diet, ID inulin diet, F fermentable diet, NF non-fermentable diet.

type *B. thetaiotaomicron* exhibited higher average daily intake of their non-fermentable ID compared to fermentable LD, predominantly driven by the cohort exposed to the non-fermentable fructan first (*Bt* NF→F) (Fig. 2d, e). They also consumed more of their non-fermentable ID when given the choice between ID and LD in an overnight feeding assay (Fig. 2f). In contrast to Fig. 1f, g, mice in the *Bt* F→NF group also showed higher average daily intake of non-fermentable ID than fermentable LD (Fig. 2e), though to a lesser degree than the NF→F group, and a similar non-significant increase in non-fermentable ID consumption overnight (Supplementary Fig. 3d). This suggests there may be some degree of post-ingestive feedback where insufficient energy extraction from non-fermentable diet leads to increased consumption.

These phenotypes were attenuated in mice colonized with *Bt^{ΔsusCD}*, which exhibited mildly increased average daily intake of ID overall, but no significant difference by diet order (Fig. 2d, e) or in overnight food choice (Fig. 2f). This suggests that bacterial fructan utilization is necessary to mediate increased host consumption of non-fermentable diet. Average daily intake was altered by colonization with *Bt^{Bo-susCD}* only in a diet order-dependent manner: *Bt^{Bo-susCD}* NF→F mice ate less of the fermentable ID compared to the non-fermentable LD (Fig. 2d, e). However, mice colonized with *Bt^{Bo-susCD}* showed no significant difference in overnight food choice (Fig. 2f). Together, these results indicate that

bacterial fructan utilization is necessary and at least partially sufficient to alter host food consumption behavior.

To confirm that bacterial fructan utilization is necessary to alter host food consumption, we next tested the effect of *Bo^{ΔsusCD}*, which cannot grow on inulin (Fig. 3a), compared to wild-type *B. ovatus*. Due to the observed baseline growth due to presumed impurities in the dietary levan and non-fermentable LD (Fig. 1a, b), we switched to a 10% cellulose diet (CD), which contains cellulose as its only carbohydrate source to serve as the *B. ovatus*-non-fermentable diet. As expected, *B. ovatus* exhibits growth in minimal media containing ID, but not CD (Fig. 3b and Supplementary Fig. 4a), which reflects a clear growth differential that was not as apparent when using LD as the non-fermentable diet (Fig. 1c). Furthermore, *B. ovatus* produced more acetate and propionate when growing on fermentable ID compared to non-fermentable CD (Supplementary Fig. 2f, g) than when growing on fermentable ID compared to non-fermentable LD (Supplementary Fig. 2d, e). Growth and SCFA production in ID were substantially decreased in the deletion strain *Bo^{ΔsusCD}* (Fig. 3b and Supplementary Figs. 2f, g and 4a), so we hypothesized that colonization with *Bo^{ΔsusCD}* would diminish the increased host non-fermentable diet intake seen with *B. ovatus* colonization. To test this, we mono-colonized mice with *B. ovatus* or *Bo^{ΔsusCD}*, fed them CD or ID in sequence for 10 days each,

and measured their food consumption (Fig. 3c). There were no differences in mouse body weight based on diet or colonization status (Supplementary Fig. 4b). There was a slight decrease in bacterial load in mice when switching from fermentable ID to non-fermentable CD, suggesting that *B. ovatus* has a growth disadvantage on cellulose but still colonizes the intestine (Supplementary Fig. 4c). Consistent with our prior observations with ID and LD (Fig. 1f, g), mice colonized with *B. ovatus* exhibited higher average daily intake of the non-fermentable CD compared to the fermentable ID (Fig. 3d), which was seen particularly in mice exposed in the NF→F diet order (Fig. 3e). This effect was diminished in mice colonized with deletion strain *Bo*[ΔsusCD] (Fig. 3d, e), again suggesting that bacterial fructan utilization mediates differential host diet intake. However, there were no differences between *B. ovatus*-colonized groups in overnight diet intake (Fig. 3f and Supplementary Fig. 4d), which differs from mice colonized with *B. thetaiotaomicron* (Fig. 2f). All together, these results support the necessity of bacterial fructan utilization in modulating host consumption of non-fermentable diet.

## Dietary SCFA supplementation is not sufficient to alter host dietary consumption

We next sought to uncover the mechanism by which bacterial fructan utilization may drive host consumption of non-fermentable fructan. The major end products of bacterial carbohydrate fermentation are SCFAs, which serve as energy sources for host intestinal epithelial cells and reduce host appetite in both mice and humans via the gut-brain axis[14–17,25–27]. We therefore hypothesized that SCFAs may mediate effects of bacterial fructan utilization on promoting host consumption of non-fermentable diets. To test this, we supplemented ID with SCFA to simulate fermentation in GF mice, which should have no gut bacterial fermentation, and measured consumption of non-fermentable CD versus "fermentable" ID containing SCFA (Supplementary Fig. 5a). Since *Bacteroides* species produce acetate and propionate but not butyrate[14], we supplemented inulin diet with sodium acetate and sodium propionate (SCFA) at physiologically relevant concentrations of 67.5 mmol/g and 25 mmol/g, respectively[28], or sodium chloride (NaCl) at 92.5 mmol/g as a sodium-matched control. There were no differences in mouse body weight based on diet condition (Supplementary Fig. 5d). We expected that mice exposed in the NF→"F" order should consume more CD than ID + SCFA, while mice in the control NF→NF group should show no difference in CD and ID + NaCl consumption. However, we observed no difference in average daily intake of CD or ID in either group (Supplementary Fig. 5b). Moreover, in the overnight assay, both groups of mice consumed more CD than ID, suggesting that NaCl and SCFA supplementation may both be mildly aversive (Supplementary Fig. 5c). These results indicate that SCFA supplementation is not sufficient to recapitulate the effect of bacterial fructan utilization on host diet consumption.

To strengthen our finding that SCFAs do not play a significant role in the effect of bacterial fructan utilization on host diet consumption, we quantified intestinal SCFA levels in a cohort of GF or mono-colonized mice that consumed either ID or LD for 7 days before sacrificing them 2 h after a controlled fasting-feeding period to synchronize temporal effects of feeding on metabolite levels[29] in cecal content (Supplementary Fig. 6a). Mice colonized with bacteria with (*Bt*, *Bo*, or *Bt*[Bo-susCD]) or without fructan utilization ability (*Bt*[ΔsusCD] or *Bo*[ΔsusCD]) exhibited higher cecal acetate and propionate levels than that of GF mice, which were close to zero (Supplementary Fig. 6b). In addition, no differences were observed between mice colonized with *Bt, Bo*, or *Bt*[Bo-susCD] consuming ID or LD, while mice colonized with *Bt*[ΔsusCD] or *Bo*[ΔsusCD] exhibited more acetate on LD than ID (Supplementary Fig. 6b). Together, these results suggest that bacterial factors other than fructan utilization contribute to overall SCFA availability in vivo (e.g., mucin degradation), which differs from in vitro where fructan utilization alone drove SCFA production (Supplementary Fig. 2d, e).

Therefore, we sought to identify other metabolites that could be differentially modified by bacterial fructan utilization by measuring polar metabolites in cecal contents of mice colonized with *Bt* or *Bt*[ΔsusCD] and fed ID or LD (Supplementary Fig. 6a). Consistent with our SCFA results (Supplementary Fig. 6b), no difference in SCFA precursors lactate and succinate were observed (Supplementary Fig. 6c). We hypothesized that metabolites influenced by bacterial fructan utilization would show different levels in mice colonized with *Bt* on non-fermentable ID and fermentable LD, while showing no difference or change in opposite direction in mice colonized with *Bt*[ΔsusCD]. Candidate metabolites that meet these criteria include acetylcholine and choline (Supplementary Fig. 6d), creatine and creatinine (Supplementary Fig. 6e), and folate (Supplementary Fig. 6f). Acetylcholine and creatine are both increased in mice colonized with *Bt* consuming non-fermentable diet compared to fermentable diet (Supplementary Fig. 6d, e), suggesting they may contribute to relatively increased consumption of non-fermentable diet. This could potentially align with known functions of acetylcholine as an excitatory neurotransmitter that regulates food intake[30–32] and creatine as a key modulator of energy utilization in neurons[33].

## Bacterial fructan utilization induces neuronal activation in feeding-related brain region after food choice

Microbial metabolites have the capacity to directly alter neural circuits underlying behavior by signaling locally to gut-innervating vagal neurons or by accessing the brain via systemic circulation and transport across the blood-brain barrier[34,35]. Acute perfusion of select microbial metabolites, including SCFAs into small intestinal lumen activates neurons in the nucleus of solitary tract (NTS), which receives direct vagal afferent innervation[36]. To test whether metabolites resulting from bacterial fructan utilization modulate gut-brain signaling via the vagal pathway, we conducted a pilot experiment to assess neuronal activation in the NTS after intestinal perfusion with sterile-filtered cecal contents collected from mice that were colonized with the different bacterial strains and fed ID or LD for 7 days (Supplementary Fig. 6g). No significant difference between perfusion of ID- or LD-derived metabolites from any group was observed (Supplementary Fig. 6h, i). Furthermore, perfusion of diet-derived metabolites from GF mice led to similar levels of neuronal activation as those from colonized mice (Supplementary Fig. 6i). Together, these results suggest that bacterial-dependent cecal metabolites produced by bacterial fructan utilization may not act through the vagal pathway to alter feeding behavior.

Neurons in arcuate nucleus of hypothalamus (ARH) are important regulators of homeostatic feeding behavior[37,38] that respond to peripheral dietary signals[39] and mediate food associations with sensory or environmental cues[38,40,41]. To test whether bacterial fructan utilization impacts neuronal activity in the ARH, we examined cFos as an early neuronal activation marker[42,43] in mice pre-exposed to the sequential feeding paradigm and then subjected to both diets to trigger acute food choice as a stimulus on day 32 (Fig. 4a). To limit potential confounding factors, mice were fasted overnight and then habituated to a sterile open field arena before exposure to both diets. Mice colonized with *Bt*, which exhibited increased consumption of non-fermentable diet (Figs. 1f–i and 2d–f), exhibited a significantly higher percentage of cFos-positive neurons in the ARH than mice colonized with *Bt*[ΔsusCD] (Fig. 4b–d), which did not exhibit such increased consumption (Fig. 2d–f). Mice colonized with *Bt*[Bo-susCD] showed a trending though non-significant increase ($p = 0.1081$) in cFos-positive neurons than mice colonized with *Bt*[ΔsusCD] (Fig. 4d), which is consistent with its more modest increase in consumption of non-fermentable diet than *Bt* (Fig. 2d–f). The major sub-populations of ARH are anorexigenic proopiomelanocortin (POMC)-expressing neurons and orexigenic agouti-related protein (AgRP)-expressing neurons[37]. In a pilot examination of *Bt* ARH sections, $83.08\% \pm 9.149\%$ of cFos+ cells were AgRP+

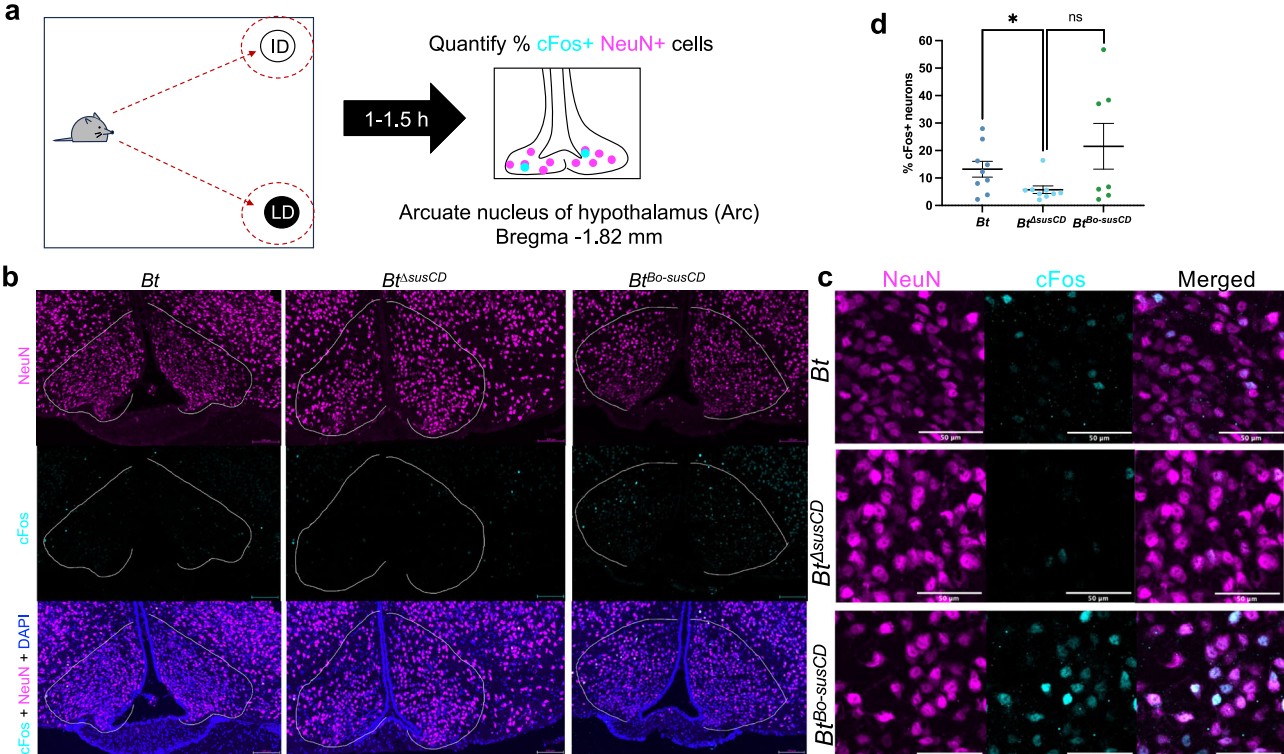

**Fig. 4 | Bacterial fructan utilization by *B. thetaiotaomicron* promotes diet choice-induced neuronal activation in the arcuate nucleus of the hypothalamus. a** Schematic of experimental set up: 1–1.5 h after the fasting-induced, mice were sacrificed, and their brains collected. Brains were sectioned coronally for the arcuate nucleus of the hypothalamus (ARH), and medial ARH sections (around Bregma −1.82) were stained for neurons with NeuN (magenta) and early activation marker, cFos (cyan). **b** Representative images of medial ARH section in mice colonized with *Bt*, *Bt^{ΔsusCD}*, and *Bt^{Bo-susCD}*. Scale bar for all panels is 100 μm. **c** Representative images of cFos-positive neurons in mice colonized with *Bt*, *Bt^{ΔsusCD}*,

and *Bt^{Bo-susCD}*. Scale bar for all panels is 50 μm. **d** Percentage of cFos-positive neurons out of total neurons in ARH of mice colonized with *Bt*, *Bt^{ΔsusCD}*, and *Bt^{Bo-susCD}*. p-values (from left to right) = 0.0394, 0.1081. Error bars (**d**) for represent mean ± SEM. Data are combined from two independent experiments with n = 9 for *Bt*, 9 for *Bt^{ΔsusCD}*, and 7 for *Bt^{Bo-susCD}*. From the first experiment (n = 3–4 per group), each dot represents the average of three technical replicates; from the second experiment (n = 4–5 per group), each dot represents one technical replicate. Unpaired t-tests (2-tailed) with Welch's correction were performed. ns = p-value > 0.10; * = p-value < 0.05. F fermentable diet, NF non-fermentable diet.

(Supplementary Fig. 7a, b), representing orexigenic neurons in the hypothalamus that promote hunger and lead to negative-valence learning[41]. No POMC signal was observed, but this is not surprising since the animals were fasted (Fig. 4a), which downregulates POMC expression[44]. Preliminary data also show no differences between mice in levels of cFos-positive neurons in the nucleus accumbens (ACB), a reward-related brain region classically implicated in food choice[6,7,9,26,27] (Supplementary Fig. 7c–e). Overall, results from this study reveal a role for bacterial fructan utilization in modulating fructan-containing diet intake and associated neuronal response in the ARH.

## Discussion

This study demonstrates that selective nutrient utilization by gut bacteria can modulate host consumption of diets containing or lacking this nutrient. We focused on bacterial metabolism of two fructans—inulin and levan—that differ only in their glycosidic linkages, which differs from prior studies that have examined microbial influences on host preference for palatable diets that differ substantially in macronutrient content[6–8]. Results from our experiments reveal that even with equivalent formulation of different fructan-containing diets, fructan-utilizing gut bacteria direct the host to increase relative consumption of diets containing the non-fermentable fructan upon sequential exposure to non-fermentable, then fermentable diets. We hypothesize that this is due to host sensing differences in energy extraction via bacterial fermentation: less energy is obtained by the host from a non-fermentable diet, so greater consumption is needed to match energy set-points determined by consumption of the fermentable diet. This

sensing may involve both microbiome-dependent post-ingestive cues, such as acetylcholine or creatine, and host satiety signaling to drive changes in diet intake.

This effect of bacterial fructan utilization on host dietary consumption was reproduced using two prevalent bacteria of the human gut microbiome—*B. thetaiotaomicron* and *B. ovatus*. Interestingly, the phenotypes observed with *B. ovatus* were consistently weaker than those observed with *B. thetaiotaomicron*, as was the efficiency of fructan utilization by *B. ovatus* compared to *B. thetaiotaomicron*. The results suggest that the strength of host diet association may depend on the efficiency of bacterial fructan degradation and amount of its fermentation products, such as the SCFA acetate, to the host. *B. thetaiotaomicron* grows robustly on its fermentable LD, reaching over $10^9$ colony forming units (CFU) by hour 12 (Fig. 1c and Supplementary Fig. 3a), while *B. ovatus* does not reach $10^9$ CFU by hour 24 when grown on its fermentable ID (Fig. 1c and Supplementary Fig. 4a), suggesting that *B. thetaiotaomicron* is more efficient than *B. ovatus* at utilizing fructans. Furthermore, *B. thetaiotaomicron* produces significantly more acetate on its fermentable LD compared to non-fermentable ID than *B. ovatus* does on its fermentable ID compared to non-fermentable LD in vitro. However, bacterial fructan utilization does not increase overall levels of SCFAs in the cecum (Supplementary Fig. 6b–c), and SCFA supplementation is not sufficient to alter host food consumption in GF mice that lack SCFA production (Supplementary Fig. 5b, c), suggesting that other metabolites generated from bacterial fructan utilization may be responsible. However, the role of SCFAs cannot be fully ruled out due to limitations of oral

supplementation compared to in vivo fermentation, such as inadequate SCFA concentration or limited effects on distal gut due to rapid absorption by upper GI tract[14], so further investigation is required.

We identified acetylcholine and creatine (Supplementary Fig. 6d, e) as candidate metabolites regulated by bacterial fructan utilization. Acetylcholine is a neurotransmitter that transduces excitatory signals between neurons and can be produced by multiple bacteria, but it cannot cross the blood-brain-barrier (BBB)[45]. However, choline can cross the BBB through specific carriers and be synthesized into acetylcholine in the brain[45]. Acetylcholine neurotransmission is involved in regulating food intake with roles varying by brain region[30–32]. Cholinergic neurons in dorsomedial hypothalamus promote food intake by enhancing GABAergic inhibitory transmission to anorexigenic POMC neurons in the arcuate nucleus of the hypothalamus[32]. In contrast, cholinergic signaling in nucleus accumbens[30] and from basal forebrain to basolateral amygdala[31] suppress appetite. Creatine moves ATP from synthesis to utilization sites within high-energy-requiring cells such as neurons; peripheral creatine crosses the BBB through a specific transporter[33]. Creatinine is a waste product of creatine and gets excreted by the kidneys into urine[46], which may explain why an inverse relationship is observed between creatine and creatinine (Supplementary Fig. 6e). Further research is warranted to test their effects on host food consumption and uncover novel molecular mechanisms for bacterial modulation of host feeding behavior.

Experiments in this study employ a reductionist approach with mono-colonized mice and engineered bacteria to demonstrate microbiome-dependent signals can modulate host food consumption in the absence of overt differences in diet palatability. In our sequential feeding paradigm, 10-day exposure to each diet was sufficient for such modulation to occur. This aligns with a previous report that mice colonized with microbiota from herbivorous wild rodents showed greater preference for diet with a higher protein to carbohydrate ratio, but only after one week[12]. We observed that the degree of modulation appears stronger depending on the order of diets that mice were exposed to: mice exposed to the non-fermentable diet first show more reliably increased consumption of non-fermentable diet than those exposed to fermentable diet first. This suggests that post-ingestive feedback from fermentable diet induces reduced consumption to a greater degree than insufficient energy extraction from non-fermentable diet induces increased consumption although it can happen (Fig. 2e).

Increased host consumption of non-fermentable diet due to bacterial nutrient utilization seems counterintuitive to the hypothesis that gut microbes may manipulate host feeding behavior to promote their own fitness[4]. However, this may apply in a complex community of microbes but not the mono-colonized gnotobiotic model in our study, highlighting a limitation of our study. Despite this limitation, investigating the effect of a single bacterium enabled us to establish a causal relationship between bacterial nutrient utilization and feeding behavior. Based on our findings, we hypothesize that bacterial fructan utilization generates energy- or satiety-related signals that the host learns to associate with fermentable diet, mainly if they are exposed to non-fermentable diet first to have a comparative baseline. One potential confound may be intestinal gas and bloating resulting from bacterial fructan fermentation, but a regular laboratory mouse diet includes ~15% soluble plant fibers[47], so we do not expect our 10% fructan diet to cause more fiber-associated discomfort than a standard diet, and we did not observe weight loss or other gross appearance or behavioral differences in mice consuming a fermentable diet. From an evolutionary perspective, increased host consumption of non-fermentable diet over fermentable diet may keep bacterial loads in check through a negative feedback loop of bacterial satiety signals by preventing a bloom of bacteria when they can utilize additional nutrient sources. This is supported by our observation that bacterial loads are not significantly

different on fermentable or non-fermentable diets (Supplementary Figs. 1c and 3c). Our finding aligns with a previous report in which intestinal *Escherichia coli* shows exponential growth in vivo in response to nutrient infusions and produces satiety proteins such as ClpB that activate anorexigenic POMC neurons in the ARH[5].

Consistent with the observation that the bacterial protein ClpB can act on ARH neurons[5], we found that bacterial fructan utilization promotes diet choice-induced neuronal activation in the ARH (Fig. 4d). We focused on ARH rather than other hypothalamic regions since bacterial acetate and ClpB act on it[5,48], and other hypothalamic regions (e.g., paraventricular nucleus and lateral hypothalamus) interact with ARH neurons to regulate feeding behavior[37,49,50]. We also found that the reward-associated ACB was not activated in response diet choice (Supplementary Fig. 7e), suggesting that the observed behavior involves homeostatic regulation, though further studies are needed to confirm and investigate other reward-associated brain regions. We hypothesize that metabolites from bacterial fructan utilization may act on the ARH to promote satiety, which over the 10-day exposure period, may lead mice to associate the fermentable diet with satiety in a diet order-dependent manner. We found that the activated cells were AgRP-positive, which may be consistent with their role in promoting negative valence learning[41]. However, future studies are needed to parse whether the neuronal activation causes or is the result of food choice behavior.

The vagus nerve plays a key role in nutrient sensing and preference[51], such as modulating nutrient preference for sugar over artificial sweetener[43,52]. It also expresses microbial metabolite receptors[53,54] and connects to the ARH[36,55] through multi-synaptic pathways, making it an ideal pathway for microbial metabolites to modulate ARH activity. Our preliminary results suggest that cecal metabolites produced by bacterial fructan utilization may not act through the vagal pathway to alter feeding behavior, but it is possible that our experiments involving intestinal perfusion of cecal filtrates require longer duration of perfusion, higher metabolite concentrations, or metabolites from other intestinal sites to elicit a vagal response. We hypothesize that bacterial metabolites may modulate host feeding behavior through multiple pathways, including vagal sensing or direct activation of brain regions via metabolites that cross the blood-brain barrier. Understanding these mechanisms could potentially inform new approaches for promoting healthier eating habits and ameliorating metabolic and eating disorders.

## Methods
### Culturing bacteria
*B. thetaiotaomicron* (VPI-5482) and *B. ovatus* (ATCC-8483) were cultured in BHIS—Brain Heart Infusion (BD) supplemented with 5 μg/ml hemin and vitamin K1 (Sigma-Aldrich)—in an anaerobic atmosphere of 85% nitrogen, 10% carbon dioxide, and 5% hydrogen. Growth curves were conducted in Salyers minimal media (MM)[56], which contains 100 mM potassium phosphate buffer (pH 7.2), 7.5 mM $(NH_4)_2SO_4$, 9.4 mM $Na_2CO_3$, 4 mM L-cysteine, 1.4 μM $FeSO_4.7(H_2O)$, 1 μg/ml vitamin K3, 5 ng/ml vitamin $B_{12}$, 1.9 μM hematin/200 μM L-histidine, 15 mM NaCl, 180 μM $CaCl_2.2H_2O$, 98 μM $MgCl_2.6H_2O$, 50 μM $MnCl_2.4H_2O$, and 42 μM $CoCl_2.6H_2O$. Fructans (fructose, inulin, levan) were sterilized by UV irradiation for 1 h and added to MM at a final concentration of 0.5%. Fructan diets (CD, ID, LD), which contain 10% fructan, were added to MM at a 5% for a final fructan concentration of 0.5%. For all growth curves, single colonies of bacteria were grown overnight in liquid BHIS, sub-cultured 1:1000 to synchronize their growth, washed with MM, and sub-cultured 1:25 into MM plus fructan or fructan diet for 24 h. Growth curves with MM plus fructan were obtained by sampling 200 μl from each tube and reading OD600 with a Synergy H1 plate reader at each timepoint. Due to the texture of fructan diets, OD600 could not be read for MM plus fructan diet. Growth curves with MM plus fructan diet were obtained by sampling 10 μl from each tube, performing serial

dilutions, and plating 10 µl of each serial dilution on BHIS plates to obtain CFU at each timepoint.

### Bacterial genetic engineering

Gene deletions for $Bt^{\Delta susCD}$ and $Bo^{\Delta susCD}$ mutants were generated using counterselectable allelic exchange with pSIE1 (Addgene plasmid #136355) as described previously[24]. Regions upstream and downstream of the *susC* and *susD* genes in the fructan PUL of *B. thetaiotaomicron* (BT1762-1763) and *B. ovatus* (BACOVA_04504-04505) were PCR amplified using custom primers ordered from ThermoFisher Scientific (primer sequences in Supplementary Table S2), ligated into the pSIE1 vector via Gibson assembly (New England Biolabs), and conjugated into *Bt* via *E. coli* S17 λ-pir and *Bo* via *E. coli* WM3064. *Bacteroides* colonies that had uptaken the vector were selected with gentamicin (200 µg/ml) and erythromycin (12.5 µg/ml), and counterselection was performed with anhydrotetracycline (100 ng/ml). After two rounds of counterselection, the desired mutants were identified by PCR screening. An isogenic strain of *B. thetaiotaomicron* that can degrade inulin but not levan ($Bt^{Bo-susCD}$) was similarly generated by counterselectable allelic exchange to replace BT1762-1763 with BACOVA_04504-04505 (Supplementary Table S2). Desired mutants then underwent counterselectable allelic exchange again to delete the levan-specific glycoside hydrolase BT1760 (Supplementary Table S2). All mutants were confirmed by PCR screening and functional test by growth curves in MM plus fructan (Supplementary Fig. 2). At 24 h after inoculation into MM plus fructan diet, 500 µl of bacterial culture was spun down at $16,000 \times g$ for 2 min to obtain cell-free supernatant. Acetate and propionate were measured in cell-free supernatant via gas chromatography performed by the UCLA Analytical Phytochemical Core.

### Sources and preparation of fructans

Fructose (F3510), chicory inulin (I2255), and pure levan (L8647) were obtained from Sigma-Aldrich. Dietary levan from Montana Biopolymers Corp. was purified by sequential water and ethanol washes to isolate higher molecular weight fructans and to remove mono- and oligosaccharides. Briefly, 200 grams of crude levan were washed with ddH$_2$O for two hours and precipitated using −20° ethanol twice. The resulting levan was then washed with 100% ethanol three times, fully dissolved in ddH$_2$O at 75 °C, filtered through Whatman paper to remove debris, and dried at 37 °C until it became a white powder.

### Mice

All experimental procedures were carried out in accordance with US NIH guidelines for the care and use of laboratory animals and approved by the UCLA Institutional Animal Care and Use Committees. Mice used for data collection were male and female GF wild-type Swiss Webster mice, at least 7–8 weeks of age. GF Swiss Webster mice were purchased from Taconic Farms and bred in flexible film isolators at the UCLA Goodman-Luskin Microbiome Center Gnotobiotics Core Facility. Mice were housed on a 12-h light-dark schedule in a temperature-controlled (22–25 °C) and humidity-controlled environment with *ad libitum* access to water and sterile "breeder" chow (Lab Diets 5K52) or experimental diets as described below.

### Gnotobiotic sequential feeding experiment

Male and female GF mice at least 6 weeks of age were transferred into flexible film isolators separated by bacterial strain and habituated to the new environment for one week. Mice were then single-housed for the duration of the experiment to measure individual diet intake. Mice were introduced to both non-fermentable and fermentable fructan diets (Inotiv-Teklad, see Supplementary Table S1 for diet formulation) −either inulin (TD.180786) and levan (TD.180785) diets, or cellulose (TD.210524) and inulin (TD.180786) diets−for 7 days. On day 8, they

were mono-colonized by one 200 µl oral gavage of turbid bacterial culture (-10$^9$ CFU/ml). Mice were then exposed to one of the two diets at a time in sequence, and diet order was counterbalanced in each cohort. Mice were exposed to the first diet for 11 days and to the second diet for 10 days. Throughout these 21 days, daily diet intake was measured every other day, mice were weighed once a week, and feces were collected and plated on BHIS supplemented with 100 µg/ml gentamicin to determine bacterial load on post-colonization days 5 and 15. On day 30, mice were presented both diets at the same time within the home cage one hour before the dark cycle for 16 h, and overnight intake of each diet was measured. Each sequential feeding experiment was performed two or three times independently, and data were aggregated.

### Non-habituation feeding experiment

Male GF mice at least 6 weeks of age were exited from flexible film isolators, housed in autoclaved cages with sterile water containing 100 µg/ml gentamicin, and mono-colonized by one 200 µl oral gavage of turbid bacterial culture (-10$^9$ CFU/ml). Conventional Swiss Webster mice at least 6 weeks of age were ordered from Taconic Farms. Mice were single-housed for the duration of the experiment to measure individual diet intake. They were given both inulin and levan diets (Inotiv-Teklad, Supplementary Table S1) for 10 days, and total intake of each diet was measured.

### Dietary SCFA supplementation experiment

GF mice at least 6 weeks of age were transferred into flexible film isolators separated by diet condition and habituated to the new environment for one week. Mice were then single-housed for the duration of the experiment to measure individual diet intake. Mice were given cellulose and inulin diet (Inotiv-Teklad, Supplementary Table S1) for 7 days. Inulin diet was supplemented with either sterile-filtered SCFA (67.5 mmol/g sodium acetate and 25 mmol/g sodium propionate[28]) or NaCl (92.5 mmol/g sodium chloride), and fresh diet was made weekly. On day 8, mice were then exposed to CD for 12 days, then inulin + SCFA or inulin + NaCl for 12 days. Throughout the experiment, daily diet intake was measured every other day, mice were weighed once a week, and feces were collected weekly and plated on Schaedler's agar to ensure sterility. On day 32, mice were presented both diets at the same time within the home cage one hour before the dark cycle for 16 h, and overnight intake of each diet was measured. This experiment was performed two times independently, and data were aggregated.

### Single diet exposure

Male GF mice at least 6 weeks of age were exited from flexible film isolators and housed in autoclaved cages with sterile water containing 100 µg/ml gentamicin. They were habituated in the new environment for 7 days and checked for sterility by plating feces on Schaedler's agar before being mono-colonized by one 200 µl oral gavage of turbid bacterial culture (-10$^9$ CFU/ml). Mice were then single-housed to measure individual diet intake and fed either inulin or levan diet for 7 days. On day 7, mice were fasted overnight and then exposed to inulin or levan diet for 1 h, during which they ate about $1 \times g$. After 2 h without diet, they were sacrificed by isofluorane anesthesia. Cecal contents were collected, immediately snap frozen in liquid nitrogen, and stored at −80° for SCFA and polar metabolite analysis and luminal perfusion experiments, described below.

### Short chain fatty acid (SCFA) measurements

The LC-MS analysis of SCFA in cecal contents was performed by the Analytical Phytochemical Core at UCLA with modifications based on previously published protocols[57,58]. 50 mg of cecal contents were homogenized in 200 µL PBS (Bead Mill, Fisher Scientific), vortexed and sonicated in ice water for 10 min. The lysate was then centrifuged at

14,000 × $g$ for 10 min at 4 °C and the supernatant was collected. 30 μl cecal lysate supernatant were mixed well with 30 μl isopropyl alcohol with isotope labelled internal standards (Acetic-d3, Propionic-d5 and Butyric-d7 5 μg/mL each), precipitated at −20 °C overnight, and centrifuged at 14,000 × $g$ for 10 min at 4 °C. The supernatant was derivatized prior to LC-MS following previously published instructions[57,58]. The LC-MS analysis was performed on an Agilent Zorbax SB-C18 2.1 × 150 mm column using the TSQ Quantum (Thermo-Finnigan) LC−ESI−MS/MS system at negative mode. Quantification was achieved by using Xcalibur data system. Calibration curves were built by fitting the analyte concentrations versus the peak area ratios of the analyte to isotope labeled internal standards. The peak area ratios target analyte to isotope-labeled internal standards in the samples were used to calculate the concentrations.

### Targeted polar metabolite analysis

Polar metabolites were extracted based on previous publications with modifications[59–61], as previously described[62]. Briefly, 50 mg of frozen cecal content was added to a pre-cooled bead beading tube containing 100 mg of 0.1 mm glass beads (Qiagen), 100 mg of 212–300 μm acid washed glass beads (Sigma-Aldrich), and 2 of 4 mm acid washed silica beads (OPS Diagnostics). 1000 μL of the extract solution (acetonitrile (ThermoScientific Chemicals, HPLC Grade)−methanol (Sigma-Aldrich, HPLC grade)−water (Fisher Chemical, HPLC Grade) = 2:2:1) was added into the tubes and homogenized in by bead beating (BioSpec Products) for 6–10 rounds of 30 s at max speed with 5 min incubations on −20 °C in between rounds until no particles were observed. Homogenate was incubated at −20 °C overnight, then centrifuged at 16,000 × $g$ for 15 min at 4 °C and stored at −20 °C until analysis. The polar metabolite separation was done on HILIC column using a UPLC system equipped with mass spectrometer (Thermo Scientific) at the UCLA Metabolomics Center as previously described[63,64]. Peaks were aligned among all samples and assigned identities using exact mass and retention time based on core's in-house database. Peaks were quantified by area under the curve integration.

### Luminal perfusion and NTS cFos immunohistochemistry

Cecal luminal contents from mono-colonized mice exposed to single diets for 7 days were pooled ($n = 5$ mice per group) and snap frozen, as described above. Cecal content was diluted to a concentration of 0.1 g/mL in sterile PBS. Samples were then centrifuged at maximum speed for 10 min to pellet out any large dietary components, and supernatants were filtered through a 0.2 μm filter. Sterile-filtered cecal contents were stored at −20° until ready to use. Conventional Swiss Webster male mice at least 6 weeks of age (Taconic Farms) were used for all perfusions. Luminal perfusion of sterile-filtered cecal contents and NTS cFos immunohistochemistry were performed as previously described[36] with modifications. Luminal perfusion: Mice were anesthetized with 5% isoflurane and maintained at 2% for the entirety of the surgery. For the inflow port, small intestine was transected at the pyloric sphincter, and a gavage needle attached to a peristaltic pump was inserted into the duodenal lumen. For the outflow port, small intestine 3 centimeters distal to the inflow port was transected. All luminal contents were flushed from the small intestine with PBS. Mice were then continually perfused with PBS at 250 μL/min flow rate for a 10-min baseline period, followed by sterile-filtered cecal content for 15 min, followed by PBS flush for 5 min. Following this perfusion, mice remained under anesthesia for one hour to allow for cFos induction. They were then sacrificed, and brains were harvested and fixed via intracardial perfusion of ice-cold PBS followed by 4% paraformaldehyde (PFA). Brains were post-fixed in 4% PFA at 4 °C for 16 h followed by an overnight incubation in 30% sucrose at 4 °C for cryoprotection, and frozen in optimal cutting temperature compound (OCT compound). Tissues were then sectioned at 30 μm into PBS containing

0.05% sodium azide. cFos immunohistochemistry (IHC): Floating sections were permeabilized in 0.5% Triton/0.05% tween-20 in PBS (PBS-TT) four times for ten minutes each, blocked with 5% normal goat serum (NGS) in PBS-TT at room temperature for two hours, and then incubated overnight at 4 °C with primary antibodies (rabbit anti-cFos (9F6) (Cell Signaling Technology 2250S) 1:500, and guinea pig anti-NeuN (Sigma-Aldrich ABN90) 1:500) in blocking solution (5% NGS + PBS-TT). The following day, sections were washed three times for ten minutes each with PBS-TT before incubating at room temperature for two hours in the dark with secondary antibodies (goat anti-rabbit 488 (ThermoFisher Scientific A-11075) 1:1000, goat anti-guinea pig 568 (ThermoFisher Scientific A-11008) 1:1000, and DAPI 1:1000) in blocking solution. They were then washed three times for ten minutes each with PBS and mounted onto slides using a thin paint brush. Confocal Imaging and Quantification Analysis: Images were obtained using a 10× water immersion objective on an upright Zeiss LSM 780 confocal microscope. Z-stacks were acquired for two technical replicates of NTS brain tissue and maximum-intensity projections were generated for subsequent analysis in ImageJ. Medial NTS sections were selected (around Bregma −7.48 mm), and ROIs were drawn based off of the Allen Brain Atlas and *The Mouse Brain in Stereotaxic Coordinates*[65,66]. All cell counts were performed by a blinded researcher. First, NeuN positive (NeuN+) cells that were each confirmed to colocalize with a DAPI nucleus were counted using the multi-point tool to obtain the total number of neurons. Subsequently, cFos+ cells were counted using the multi-point tool by confirming colocalization of a cFos immunofluorescence signal with NeuN and DAPI. For each image, the total number of cFos+ neurons were divided by the total number of NeuN+ cells to obtain the percentage of cFos+ neurons. Finally, the percentage of cFos+ neurons for all technical replicates of NTS slices per animal were averaged to obtain a biological $n = 1$ and find the overall percentage of cFos+ neurons.

### cFos immunohistochemistry in ARH and ACB

Behavior stimuli: Mono-colonized mice that had gone through the sequential feeding experiment were subjected to both diets on day 32 to trigger acute food choice as a stimulus. Mice were fasted overnight and then habituated to a sterile open field arena for 10 min before exposure to both diets for 20 min. Following a one hour rest period for cFos induction, animals were sacrificed by isoflurane, and tissues were fixed via intracardial perfusion of ice-cold PBS followed by 4% PFA. Brains were then post-fixed in 4% PFA at 4 °C for 16 h, followed by incubation in 30% sucrose at 4 °C for cryoprotection, and frozen in OCT compound. Tissues were then sectioned at 20 μm and mounted on microscope slides. cFos immunohistochemistry (IHC): IHC was performed as described previously[36]. Briefly, slides were permeabilized in 0.5% Triton/0.05% Tween-20 in PBS (PBS-TT) and blocked with 5% NGS in PBS-TT at room temperature for two hours. The tissue sections were then incubated overnight at 4 °C with primary antibodies (rabbit anti-cFos (9F6) (Cell Signaling Technologies 2250S) 1:500, and guinea pig anti-NeuN (Sigma-Aldrich ABN90) 1:500 or guinea pig anti-AgRP (ThermoFisher Scientific PA1-18414) 1:200) in blocking solution (5% NGS + PBS-TT). The following day, slides were washed with PBS-TT before incubating with secondary antibodies (goat anti-rabbit 488 (ThermoFisher Scientific A-11075) 1:1000, goat anti-guinea pig 568 (ThermoFisher Scientific A-11008) 1:1000, and DAPI (ThermoFisher) 1:1000) in blocking solution at room temperature in the dark for two hours. Confocal Imaging and Quantification Analysis: Images were obtained and analyzed as described previously[36]. Briefly, technical replicates of ARH and ACB were imaged using a 20× air objective (NA 0.8) on an upright Zeiss LSM 780 confocal microscope. Medial ACB (around Bregma +1.18 mm) and medial ARH sections (around Bregma −1.82 mm) were selected, and ROI's were drawn based off of the Allen Mouse Brain Atlas and *The Mouse Brain in Stereotaxic Coordinates*[65,66]. All cell counts were performed by a

blinded researcher and technical replicates were averaged, as described above.

## Quantification and statistical analysis

Statistical analysis was performed using Prism software version 10.1.0 (GraphPad). Data were assessed for normal distribution and plotted in the figures as mean ± SEM. For each figure legend, $n$ = the number of independent biological replicates. Differences among >2 groups with 2 variables were assessed using two-way ANOVA and matching across rows with Sidak's corrections. Differences between two treatment groups were assessed using unpaired two-tailed Student $t$-test. Significant differences emerging from the above tests are indicated in the figures by ns = $p$-value > 0.10, * = $p < 0.05$, ** = $p < 0.01$, *** = $p < 0.001$, **** = $p < 0.0001$.

## Data availability

All data supporting the findings of this study are available within the paper and its Supplementary Information. Source data are provided in the Supplementary Information. Source data are provided with this paper.

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

## Acknowledgements

We thank members of the Hsiao laboratory for their guidance and review of the manuscript, especially Dr. Kelly Jameson and Dr. Stephanie Orchanian for technical guidance; members of the UCLA Goodman-Luskin Microbiome Center Gnotobiotics Core Facility for protocol development and technical support; Dr. Jieping Yang, Dr. Ru-Po Lee, and Dr. Qing-Yi Lu of the UCLA Analytical Phytochemical Core for measuring SCFAs; Dr. Johanna ten Hoeve-Scott at the UCLA Metabolomics Core for polar metabolite analysis; Dr. Karen Reue and Dr. Laurent Vergnes for conducting pilot investigations of mitochondrial metabolism; Dr. Andrew Goodman for providing the pSIE1 plasmid; and Dr. Joan Combie (Montana Biopolymers Corp.) for producing levan and advising on its purification. This work was supported by funds from a UCLA-Caltech Medical Scientist Training Program T32 (NIGMS 3T32GM008042-35S1) and UCLA Whitcome Fellowship to K.B.Y. and W.M. Keck Foundation grant to E.Y.H. E.Y.H. is a New York Stem Cell Foundation—Robertson Investigator. This research was supported in part by the New York Stem Cell Foundation.

## Author contributions

K.B.Y., C.S., E.O., A.C., J.P., A.N., S.A.K., G.R.L. and A.L. performed the experiments. K.B.Y., C.S., E.O., A.C., A.V., A.R., D.F., F.C. and A.N. analyzed the data. E.O. and J.B.L. provided key technical guidance and resources. K.B.Y. and E.Y.H. designed the study and wrote the manuscript. All authors discussed the results and commented on the manuscript.

## Competing interests

The authors declare no competing interests.
