## [Transparent Peer Review file · Nature Communications]

Complex carbohydrate utilization by gut bacteria modulates host food consumption

Corresponding Author: Professor Elaine Hsiao

Version 0:

Reviewer comments:

Reviewer #1

(Remarks to the Author)

In this manuscript by Yu et al., the authors further explore how gut microbial metabolism modulates host feeding behavior, specifically focusing on the ability of select *Bacteroides* species to ferment fructans of different glycosidic linkages. The authors previously demonstrated that mice consume more of the diet containing the non-fermentable fructan relative to the one metabolized by their gut symbiont. In this revised version, the authors address prior concerns by shifting their emphasis from "food preference" to "food consumption," offering additional data and improved mechanistic insight.

Overall, I found the revised manuscript much improved and appreciate the authors' thoughtful responses to reviewer and editor feedback. The main conclusions are now more cautiously framed, and several additional datasets, including daily intake curves, revised statistical treatment, and clarifications about brain activation, improve confidence in the interpretation. The inclusion of extended metabolite and SCFA profiling, along with acknowledgement of limitations in the vagal signaling assays, is commendable.

As Reviewer 1 on the initial submission, I am now satisfied with the changes made in response to the editorial and reviewer queries. I only have a few comments, none of which should prevent publication.

MAJOR IMPROVEMENTS

1. Reviewer 2's concerns appear to be addressed adequately. The language has been revised to avoid overinterpreting the behavior as preference or learning. The hypothesis that animals modulate food consumption in response to energy extraction (rather than palatability or reward) is now clearly articulated and well supported. I appreciate that the authors reframed their interpretation in terms of post-ingestive feedback and clarified the limitations of their approach.
2. The authors now include daily food intake time courses, which convincingly show that intake reduction occurs after exposure to the fermentable diet, consistent with an energy-driven satiety mechanism rather than innate preference. This directly addresses Reviewer 2's request and strengthens the manuscript considerably.
3. Terminology: The manuscript now avoids the term "preference" outside of citations, which I strongly support. "Consumption" or "intake" is used consistently and appropriately.
4. Figures: The added clarification in Extended Data Fig. 5 and 6 concerning SCFA supplementation and metabolite profiles (e.g. acetylcholine, creatine) is useful, even if exploratory. The acknowledgment of absorption dynamics for SCFAs is also important.
5. cFos quantification: The authors now clearly state the limits of their design and refrain from overinterpreting the ARH data. The increased N in Fig. 4d helps support their claims. They also acknowledge the possibility of additional brain regions being involved.

MINOR SUGGESTIONS

Statistical analysis is broadly sound, with appropriate use of 2-way ANOVA and Sidak correction. Sample sizes have been increased in critical behavioral and brain activation experiments (notably in Figure 2 and 4). The authors also transparently refer to neuronal assays as "pilot experiments" when sample size remains low. That said, future work would benefit from mixed models for repeated measures or sections nested within animals (particularly for cFos analyses).

Statistical clarity: P-values are generally well reported. I still recommend removing references to "trending" significance (e.g., $p = 0.109$) or, at least, explicitly noting these are not statistically significant.

This is a well-executed and novel study that reveals a microbiota-dependent modulation of host food intake behavior, likely

via energy-sensing mechanisms rather than innate preference. The authors have responded thoroughly and transparently to prior feedback, and the statistical treatment is now sound. I support publication of this revised version.

Reviewer #3

(Remarks to the Author)

In this last version of the manuscript, the authors have edited and improved the wording used in the manuscript thereby avoiding misuse of wording and overinterpretation. I think the message is more robust and accurate now.

I just have a minor comment regarding the line 166 that has been added in this new version of the manuscript:

"Together, these results show that gut microbial fructan utilization leads to a change in food consumption behavior: mice colonized with *B. thetaiotaomicron* consumed relatively less of the fermentable ID than non-fermentable LD when exposed in NFàF sequence (Fig. 1f-i), which may be driven by post-ingestive feedback of more energy extracted from the fermentable diet, despite a baseline preference for LD diet (Extended Data Fig. 1f). This aligns with our hypothesis that host sensing of differential energy extraction by bacteria would alter host fructan diet consumption."

I guess that the fermentable diet for *B. thetaiotaomicron* should be noted LD whereas the non-fermentable should be noted ID

I don't have any additional comment.

REVIEWERS' COMMENTS

Reviewer #1 (Remarks to the Author):

In this manuscript by Yu et al., the authors further explore how gut microbial metabolism modulates host feeding behavior, specifically focusing on the ability of select *Bacteroides* species to ferment fructans of different glycosidic linkages. The authors previously demonstrated that mice consume more of the diet containing the non-fermentable fructan relative to the one metabolized by their gut symbiont. In this revised version, the authors address prior concerns by shifting their emphasis from "food preference" to "food consumption," offering additional data and improved mechanistic insight.

Overall, I found the revised manuscript much improved and appreciate the authors' thoughtful responses to reviewer and editor feedback. The main conclusions are now more cautiously framed, and several additional datasets, including daily intake curves, revised statistical treatment, and clarifications about brain activation, improve confidence in the interpretation. The inclusion of extended metabolite and SCFA profiling, along with acknowledgement of limitations in the vagal signaling assays, is commendable.

As Reviewer 1 on the initial submission, I am now satisfied with the changes made in response to the editorial and reviewer queries. I only have a few comments, none of which should prevent publication.

MAJOR IMPROVEMENTS

1. Reviewer 2's concerns appear to be addressed adequately. The language has been revised to avoid overinterpreting the behavior as preference or learning. The hypothesis that animals modulate food consumption in response to energy extraction (rather than palatability or reward) is now clearly articulated and well supported. I appreciate that the authors reframed their interpretation in terms of post-ingestive feedback and clarified the limitations of their approach.
2. The authors now include daily food intake time courses, which convincingly show that intake reduction occurs after exposure to the fermentable diet, consistent with an energy-driven satiety mechanism rather than innate preference. This directly addresses Reviewer 2's request and strengthens the manuscript considerably.
3. Terminology: The manuscript now avoids the term "preference" outside of citations,

which I strongly support. “Consumption” or “intake” is used consistently and appropriately.

4. Figures: The added clarification in Extended Data Fig. 5 and 6 concerning SCFA supplementation and metabolite profiles (e.g. acetylcholine, creatine) is useful, even if exploratory. The acknowledgment of absorption dynamics for SCFAs is also important.

5. cFos quantification: The authors now clearly state the limits of their design and refrain from overinterpreting the ARH data. The increased N in Fig. 4d helps support their claims. They also acknowledge the possibility of additional brain regions being involved.

We thank the Reviewer for acknowledging our efforts to further clarify our findings and limitations.

MINOR SUGGESTIONS

Statistical analysis is broadly sound, with appropriate use of 2-way ANOVA and Sidak correction. Sample sizes have been increased in critical behavioral and brain activation experiments (notably in Figure 2 and 4). The authors also transparently refer to neuronal assays as “pilot experiments” when sample size remains low. That said, future work would benefit from mixed models for repeated measures or sections nested within animals (particularly for cFos analyses).

Statistical clarity: P-values are generally well reported. I still recommend removing references to “trending” significance (e.g., $p = 0.109$) or, at least, explicitly noting these are not statistically significant.

This point is well-taken and we have added “trending though non-significant” for one instance and removed the other.

This is a well-executed and novel study that reveals a microbiota-dependent modulation of host food intake behavior, likely via energy-sensing mechanisms rather than innate preference. The authors have responded thoroughly and transparently to prior feedback, and the statistical treatment is now sound. I support publication of this revised version.

We thank the Reviewer for acknowledging the novelty of our study and our revisions, and for carefully reviewing the manuscript to offer further suggestions.

Reviewer #3 (Remarks to the Author):

In this last version of the manuscript, the authors have edited and improved the wording used in the manuscript thereby avoiding misuse of wording and overinterpretation. I think the message is more robust and accurate now.

We thank the Reviewer for acknowledging our revisions and for the improvements they had suggested.

I just have a minor comment regarding the line 166 that has been added in this new version of the manuscript:

"Together, these results show that gut microbial fructan utilization leads to a change in food consumption behavior: mice colonized with *B. thetaiotaomicron* consumed relatively less of the fermentable ID than non-fermentable LD when exposed in NF→F sequence (Fig. 1f-i), which may be driven by post-ingestive feedback of more energy extracted from the fermentable diet, despite a baseline preference for LD diet (Extended Data Fig. 1f). This aligns with our hypothesis that host sensing of differential energy extraction by bacteria would alter host fructan diet consumption."

I guess that the fermentable diet for *B. thetaiotaomicron* should be noted LD whereas the non-fermentable should be noted ID

I don't have any additional comment.

Thank you for the careful review. We have corrected this sentence now (Line 165):
"Together, these results show that gut microbial fructan utilization leads to a change in food consumption behavior: mice colonized with *B. thetaiotaomicron* consumed relatively less of the fermentable LD than non-fermentable ID when exposed in NF→F sequence..."